

# Protective effects of the extracts of *Barringtonia racemosa* shoots against oxidative damage in HepG2 cells

Kin Weng Kong[1], Sarni Mat-Junit[1], Norhaniza Aminudin[2],
Fouad Abdulrahman Hassan[3], Amin Ismail[3] and Azlina Abdul Aziz[1]

[1] Department of Molecular Medicine, Faculty of Medicine, University of Malaya, Kuala Lumpur, Malaysia
[2] Institute of Biological Sciences, Faculty of Science, University of Malaya, Kuala Lumpur, Malaysia
[3] Department of Nutrition and Dietetics, Faculty of Medicine and Health Sciences, Universiti Putra Malaysia, Serdang, Selangor, Malaysia

## ABSTRACT

*Barringtonia racemosa* is a tropical plant with medicinal values. In this study, the ability of the water extracts of the leaf (BLE) and stem (BSE) from the shoots to protect HepG2 cells against oxidative damage was studied. Five major polyphenolic compounds consisting of gallic acid, ellagic acid, protocatechuic acid, quercetin and kaempferol were identified using HPLC-DAD and ESI-MS. Cell viability assay revealed that BLE and BSE were non-cytotoxic (cell viabilities >80%) at concentration less than 250 μg/ml and 500 μg/ml, respectively. BLE and BSE improved cellular antioxidant status measured by FRAP assay and protected HepG2 cells against $H_2O_2$-induced cytotoxicity. The extracts also inhibited lipid peroxidation in HepG2 cells as well as the production of reactive oxygen species. BLE and BSE could also suppress the activities of superoxide dismutase and catalase during oxidative stress. The shoots of *B. racemosa* can be an alternative bioactive ingredient in the prevention of oxidative damage.

## INTRODUCTION

Oxidative stress is attributed to physiological imbalance between the production of reactive oxygen species (ROS) and antioxidant defense capability, in favour of the former (*Choi et al., 2010*). It is a crucial factor that contributes to aging and multiple degenerative diseases owing to the alteration of biological molecules such as DNA, proteins and lipids (*Yoshihara, Fujiwara & Suzuki, 2010*). Endogenous and exogenous antioxidants are the important candidates for maintaining the oxidative balance of human physiology and diminishing the impact of ROS.

Fruits and vegetables containing phytochemicals such as polyphenols and carotenoids are good examples of exogenous antioxidants that can help in reducing oxidative stress (*Alía et al., 2006a*; *Kong et al., 2010*). Many of these bioactive compounds do not only serve as exogenous antioxidants, but also offer indirect protection via the regulation of the activities of antioxidant enzymes such as catalase, superoxide dismutase and glutathione peroxidase (*Alía et al., 2006a*).

Corresponding author
Azlina Abdul Aziz,
azlina_aziz@um.edu.my

*Barringtonia racemosa* (L.) Spreng is a tropical or subtropical plant belonging to the Lecythidaceae family. In Malaysia, the shoots of this wildly grown plant are usually consumed as salad, either fresh or blanched (*Lim, 2012*). Previous studies by our group using chemical and biological antioxidant assays demonstrated that the water extracts of *B. racemosa* shoots had excellent antioxidant properties as a result of their high amounts of polyphenols (*Kong et al., 2012*). The prominent polyphenolic compounds identified in the *B. racemosa* extracts were gallic acid, ellagic acid and quercetin (*Kong et al., 2014*). Antioxidant analyses of *B. racemosa* using cellular model has never been conducted and information obtained from such study can provide useful data particularly with regards to their ability to protect cells against oxidative damage.

Hepatocellular carcinoma cells, HepG2, are a well established cell line and a reliable model in studying the antioxidant effects of dietary compounds (*Alía et al., 2006b*). Phenolic acids and flavonoids from plants are metabolised by the liver after absorption, mainly, in the small intestine (*Martín et al., 2008*). In this study, HepG2 cells were used as a cellular model to further investigate the effects of the water extracts of *B. racemosa* on the antioxidant defense systems as well as their ability to protect the cells against oxidative damage. Data obtained will provide further evidence to support the biological action of *B. racemosa* extracts, particularly as a potent source of antioxidative agents.

# MATERIALS AND METHODS

## Analytical reagents and chemicals

HPLC grade or analytical grade solvents and chemicals were purchased from the general suppliers. Polyphenolic standards used were of HPLC grade (purity >95%) including gallic acid, protocatechuic acid, ellagic acid, quercetin and kaempferol. These polyphenolic standards were purchased from Sigma-Aldrich Chemical Co. (St. Louis, MO, USA).

## Sample preparation and extraction

The shoots of *B. racemosa* were obtained from the state of Kedah, located in northern Peninsular Malaysia. The voucher specimen (KLU48175) of the sample was deposited in the Herbarium of Rimba Ilmu, University of Malaya. The shoots were separated into two parts; the leaf and the stem portions. The lyophilised samples were ground and sieved via a 1 mm mesh. Plant extraction was performed following the method of *Kong et al. (2012)*. Briefly, 2 g of dried sample was extracted with 40 ml of water at 30 °C for 24 h. Following centrifugation, the resulting supernatant was subjected to lyophilisation and re-dissolved in water to give the *B. racemosa* leaf (BLE) and stem (BSE) extracts. The extracts were passed through a sterilised 0.22 µm syringe filter before the cell culture treatments. Gallic acid standard was used for comparison in the cell-based assays, as it is one of the major polyphenols found in *B. racemosa*.

## Analysis of polyphenols in *B. racemosa* using HPLC-DAD and ESI-MS

Lyophilised extracts (10 mg) were hydrolysed in 2 ml of 1.2 N HCl containing 20 mM DETC sodium salt in a hydrolysis vial. The hydrolysis was conducted in a heating module at 90 °C for 2 h. The hydrolysate was centrifuged and the supernatant filtered via 0.20 µm

PTFE membrane filters prior to chromatographic analysis. Hydrolysis was performed in order to release the free polyphenols (aglycone) from the conjugated forms, hence allowing easier identification of the polyphenols in the samples. High performance liquid chromatography-diode array detector (HPLC-DAD) (Agilent 1100, Santa Clara, USA) and electrospray ionisation-mass spectrometry (ESI-MS) analyses were conducted following the method of *Hassan et al. (2011)*. For the HPLC analyses, the stationary phase comprised of a reversed-phased Lichrospher $C_{18}$ column (250 mm × 4 mm, i.d. 5 μm, Merck, Germany), at a temperature of 30 °C. Gradient elution system was applied using 0.2% acetic acid (solvent A) and methanol (solvent B) with a flow rate of 0.8 ml/min. A linear gradient system was employed for the separation: 5–90% B in 20 min, 90% B in 5 min, 90–5% B in 5 min. The polyphenolic compounds were detected by DAD at 280 nm. Identification of polyphenolic compounds was done by comparing the retention times with that of the authentic standards.

Polyphenolic compounds detected in the extracts were further confirmed using ESI-LC-MS using an Applied TSQ Quantum Ultra-LCMS system (Thermo Fisher, USA). Both negative and positive modes electrospray ionisation (ESI±) of the mass spectrometer was applied. The capillary temperature was set at 270 °C and the spray voltage was 3,500 V. The sheath/auxiliary/sweep gas was 99% pure nitrogen, and the sheath gas pressure was 30 psi with 5 psi for the auxiliary gas pressure. The injection volume was 10 μl and flush speed was 100 μl/s. The mass to charge ratio ($m/z$) was obtained through the full scan mass in the range of $m/z$ 100–800. The identified polyphenolic compounds were confirmed by comparing the $m/z$ with their molecular weight and the $m/z$ of the authentic standards.

## Cell culture

Human hepatoma HepG2 cell line was obtained from the American Type Culture Collection (ATCC) (Manassas, VA, USA). Cells were cultivated in DMEM with 2.0 g/l sodium bicarbonate, antibiotics (100 units of penicillin/ml and 100 μg of streptomycin/ml) and 10% fetal bovine serum (FBS). Cells were maintained in a humidified atmosphere of 5% $CO_2$ at 37 °C.

## Cytotoxicity effects

Cell viability was measured using 3-(4,5-dimethylthiazole-2yl)-2,5-diphenyl tetrazolium bromide (MTT) assay (*Mosmann, 1983*). Briefly, HepG2 cells were plated at 5 × 10³ cells per well in 96 well plates supplemented with 100 μl DMEM growth medium. After stabilising the cells, the culture medium was replaced with 200 μl of medium containing different concentrations (0–500 μg/ml) of BLE, BSE and gallic acid. Cells were incubated for 48 h at 37 °C with 5% $CO_2$. After 48 h, 20 μl of MTT reagent (5 mg/ml, prepared in phosphate buffered saline, PBS) was added to the medium. The MTT reagent was removed after 4 h, and formazan crystals formed were dissolved in 100 μl of DMSO. The absorbance was read at 570 nm (Bio-Rad Model 680 microplate reader, California, USA). Inhibition of cell growth by the sample was calculated and expressed as percentage of cell viability. A non-toxic sample concentration (>90% cell viability) was selected for further analyses.

## Cellular antioxidant status

Cellular antioxidant status was determined using the ferric reducing power (FRAP) assay (*Benzie & Strain, 1996*). HepG2 cells were plated at $5 \times 10^3$ cells per well in 100 µl DMEM in a 96 well plate. Following stabilisation, the cells were treated with 200 µl of BLE and BSE using two levels of concentrations prepared in DMEM: high concentrations (5, 10, 20 µg/ml) and low concentrations (0.5, 1, 2 µg/ml); or gallic acid: high concentrations (5, 10, 20 µM) and low concentrations (0.5, 1, 2 µM). The concentration of gallic acid was estimated based on the total polyphenolic content reported in our previous study (*Kong et al., 2012*). After 24 h of incubation, cells were washed 3 times with PBS and 100 µl of 25 mM Tris–HCl (pH 7.4) was added to the medium. The plate was ultrasonicated for 5 min to induce cell rupture. Freshly prepared FRAP reagent (300 mM acetate buffer, 10 mM ferric-tripyridyl triazine, 20 mM iron (III) chloride, 10:1:1) was added and incubated at 37 °C for 30 min. The absorbance was read at 595 nm. Iron sulphate ($FeSO_4$) at a concentration range of 0–1,000 µM was used as standard and analysed as above. Results were expressed as µM of ferrous ion ($Fe^{2+}$).

## Cytoprotective effects

The cytoprotective effects of BLE, BSE and gallic acid were determined by a modified method of *Kong et al. (2010)*. HepG2 cells were seeded in 96 well plates at $5 \times 10^3$ cells per well. The cells were supplemented with 100 µl DMEM for 24 h at 37 °C with 5% $CO_2$ in a humidified atmosphere. Then, cells were pre-incubated with BLE, BSE (0–20 µg/ml) or gallic acid (0–20 µM) for 24 h. After three washes with PBS, 200 µl of $H_2O_2$ (300 µM) solution was added to induce cellular damage or cell death. After 24 h, cell viability was measured using MTT assay as previously described. Positive and negative controls included cells treated with $H_2O_2$ or medium alone, respectively. The cytoprotective effect was expressed as the percentage of viable cells following treatments.

## Analysis of cellular reactive oxygen species (ROS)

The changes of intracellular ROS levels were measured accordingly based on a modified method of *Choi et al. (2010)*. HepG2 cells ($5 \times 10^3$ cells per well) were plated into 96-well plates and allowed to stabilise for 24 h before being pre-treated with BLE, BSE (0–20 µg/ml) or gallic acid (0–20 µM) for 24 h. After three washes with PBS, the cells were incubated in the dark with 100 µM 2,7-dichlorodihydrofluorescein diacetate (DCFH-DA), prepared in serum-free media, for 30 min, at 37 °C. Subsequently, cells were washed twice with PBS and incubated with 1 mM $H_2O_2$ for 1 h. Fluorescence reading was taken with the excitation and emission wavelengths set at 485 nm and 530 nm (Varian Cary Eclipse Fluorescence Spectrophotometer, USA), respectively. Positive and negative controls consisted of cells treated with $H_2O_2$ but without sample treatment and cells containing medium alone, respectively. Results were expressed as relative fluorescence unit.

## Analysis of lipid peroxidation

HepG2 cells were plated at $1.5 \times 10^5$ cells per well in 6 well plates and allowed to stabilise for 24 h prior to treatment with 2 ml of BLE, BSE (5–20 µg/ml) or gallic acid (5–20 µM). After incubation for 24 h, lipid peroxidation was induced with 2 ml of $H_2O_2$ (1 mM) for

1 h (*Puiggròs et al., 2005*). Cells were then gently washed thrice with PBS and harvested in 1.5 ml of PBS by scraping. Following centrifugation, the pellet was re-suspended in 100 $\mu$l of 25 mM Tris–HCl buffer (pH 7.4) and subjected to ultrasonication for 5 min. Protein content of the cell suspension was measured using bovine serum albumin as standard (*Bradford, 1976*).

The extent of lipid peroxidation was estimated by measuring levels of malondialdehyde (MDA) using the thiobarbituric acid reactive substances (TBARS) assay (*Buege & Aust, 1978*). Ninety microlitres of the reaction mixture was mixed with 180 $\mu$l of thiobarbituric acid (0.37%), trichloroacetic acid (15%), and hydrochloric acid (0.25 N) at a ratio of 1:1:1. The mixture was heated in a 90 °C water bath for 20 min and cooled at room temperature for 10 min. Following centrifugation, absorbance of the supernatant was measured at 532 nm. Positive and negative controls consisted of cells treated with $H_2O_2$ and medium alone, respectively. A standard calibration curve was prepared from 1,1,3,3-tetraethoxypropane (TEP) (0–0.02 $\mu$mol/ml), a commercial form of MDA. Results were expressed as nmol MDA equivalents/$\mu$g protein.

## Analysis of cellular antioxidant enzyme activities

HepG2 cells ($1.5 \times 10^5$ cells per well) were plated into 6-well plates and stabilised for 24 h prior to treatment with 2 ml of BLE, BSE (5–20 $\mu$g/ml) or gallic acid (5–20 $\mu$M). After the treatment, the cells were subjected to induction of oxidative stress by incubating the cells for 1 h with 2 ml of $H_2O_2$ (1 mM). After the incubation, cells were washed three times with PBS and harvested by scraping. The cells were ultrasonicated for 5 min in 0.2 ml of PBS containing 25 mM Tris–HCl (pH 7.4). Protein content of the cell suspension was measured (*Bradford, 1976*). The cells were subsequently centrifuged and the supernatant was kept at −20 °C until further analysis. Positive and negative controls consisted of cells treated with $H_2O_2$ and medium alone, respectively. Superoxide dismutase (SOD) and catalase (CAT) activities were determined using assay kits following the manufacturer's instructions.

## Superoxide dismutase activity

SOD activity was conducted according to the manufacturer's instructions (Cayman, USA). The capability of SOD to cause dismutation of superoxide anion radicals ($O_2^{-\bullet}$) generated from xanthine oxidase and hypoxanthine was measured. A diluted tetrazolium salt was used as radical detector. One unit (U) of SOD is defined as the amount of enzyme needed to produce 50% dismutation of $O_2^{-\bullet}$. The SOD activity was expressed as U/mg protein.

## Catalase activity

CAT activity was assayed according to the manufacturer's instructions (Cayman, USA). This assay is based on the peroxidatic activity caused by CAT on the reaction between methanol and $H_2O_2$ that forms formaldehyde and water. Formaldehyde formed can be measured using 4-amino-3hydrazino-5-mercapto-1,2,4-triazole (purpald) as chromogen. One unit (U) of CAT is defined as the amount of enzyme that catalyses the formation of 1 nmol of formaldehyde per minute at 25 °C. CAT activity was expressed as U/mg protein.

## Statistical analysis

All data were expressed as mean $\pm$ standard error of means (SEM) of three independent experiments. Data were statistically analysed using the SPSS statistical software version 15 (SPSS Inc, Chicago, Illinois, USA). One-way analysis of variance (ANOVA) and Fisher's least significant difference test were used to compare means among the groups. Independent $t$-test was used for comparison between groups. The level of significance was set at $p < 0.05$.

# RESULTS AND DISCUSSION

## HPLC-DAD and ESI-MS analyses of polyphenols in *B. racemosa*

HPLC-ESI-MS is an effective tool for identification and characterisation of polyphenolic compounds (*Hassan et al., 2011*). MS ionises polyphenolic compounds to their charged forms, from which their mass to charge ratios ($m/z$) can be determined. HPLC analysis of the shoots of *B. racemosa* identified the presence of gallic acid, protocatechuic acid, ellagic acid, quercetin and kaempferol in BLE whereas only gallic acid, protocatechuic acid and ellagic acid were detected in BSE. The presence of these polyphenols in the plant extracts were further confirmed using ESI-MS (Figs. 1A and 1B). Figures 1C–1G shows the mass to charge ratio ($m/z$) of the polyphenols detected using ESI-MS analyses. The polyphenolic compounds identified in BLE and BSE were in agreement with our previous study, analysed using ultra high performance liquid chromatography (*Kong et al., 2014*).

In mass spectrometry analyses, gallic acid and kaempferol were detected in ESI ($-$) modes, with [M$-$H]$^-$ peak observed at $m/z$ 168.96 for gallic acid (Fig. 1C) and $m/z$ 284.98 for kaempferol (Fig. 1G). Protocatechuic acid was monitored in ESI ($+$) mode, with [M$+$H]$^+$ peak observed at $m/z$ 155.21 (Fig. 1D). On the other hand, both ESI negative and positive modes were able to detect ellagic acid and quercetin in the samples, however ESI ($-$) mode was selected due to better sensitivity and lower background noise. The [M$-$H]$^-$ peak of ellagic acid and quercetin were observed at $m/z$ 301.05 and 300.95, respectively (Figs. 1E and 1F). The polyphenols detected in *B. racemosa* were confirmed as they were in agreement with the $m/z$ of their standard and molecular weight.

## Cytotoxicity effects

Toxicity study was conducted to ascertain that the extracts were safe for the proposed treatments on the HepG2 cells. High dosage of dietary compounds could be toxic or mutagenic in cell culture system, producing adverse metabolic reactions in mammals (*Alía et al., 2006a*). Hence, the direct effects of BLE, BSE and gallic acid on cell viability of HepG2 cells at different dosages were investigated (Table 1).

BLE and BSE were relatively non-toxic to HepG2 cells at concentrations less than 200 μg/ml, with cell viability more than 90%. Moreover, at concentrations less than 100 μg/ml, BSE showed higher cell viability (>100%) indicating that the extract can stimulate cell growth. Indeed, previous studies have reported low toxicity of the water extracts of plants such as dandelion root and common sage on HepG2 cells (*Lima et al., 2007*; *You et al., 2010*). The cytotoxicity of plant extracts are highly dependent on their concentration, bioavailability and together with the complex interaction among the phytochemicals, may either cause cell damage or be protective against it

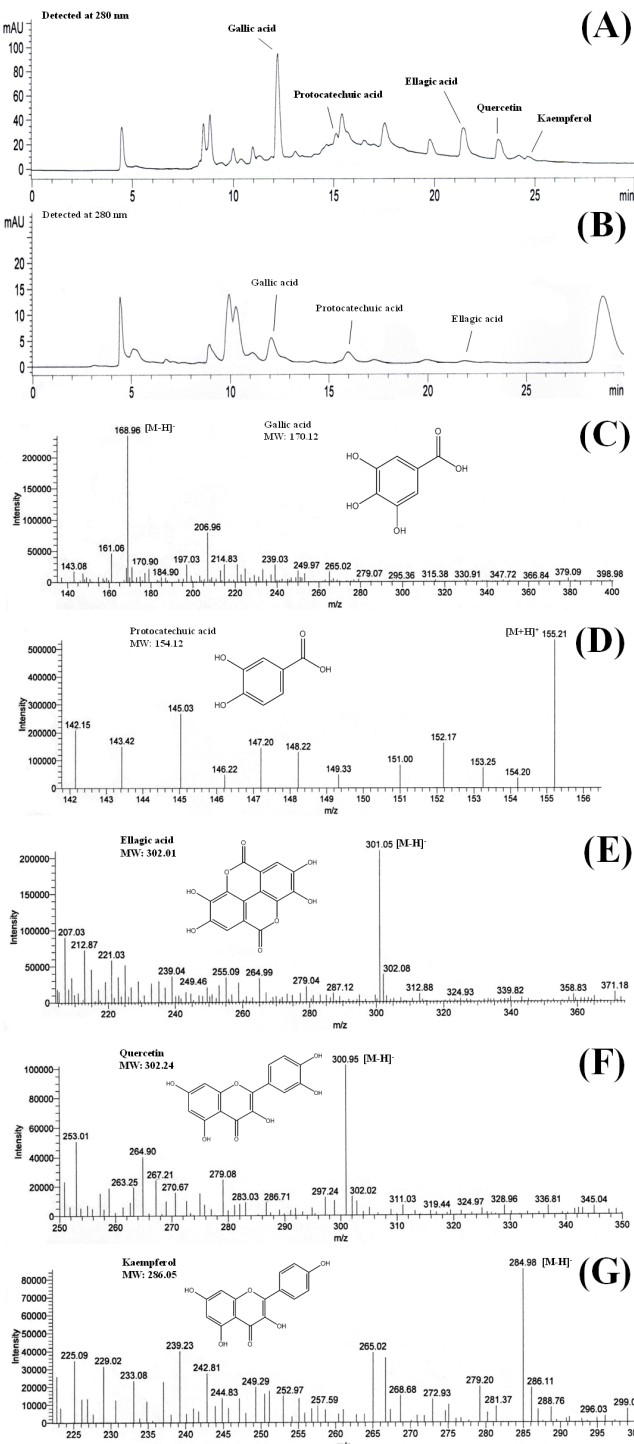

**Figure 1** **Chromatograms of (A) BLE and (B) BSE and the mass to charge ratio ($m/z$) of (C) gallic acid, (D) protocatechuic acid, (E) ellagic acid, (F) quercetin and (G) kaempferol.** The chromatograms were obtained from HPLC-DAD analyses while the $m/z$ was obtained from ESI-MS analyses. $[M + H]^+$ and $[M-H]^-$ are the ions of the detected compounds obtained from the negative and positive full scan modes. BLE, Leaf water extract of *B. racemosa*; BSE, Stem water extract of *B. racemosa*.

**Table 1** The effects of gallic acid, BLE and BSE on cell viability of HepG2 cells.

| Treatment (µg/ml) | Cell viability (%) | | |
|---|---|---|---|
| | GA | BLE | BSE |
| 0.00 | $100.00 \pm 0.00$ | $100.00 \pm 0.00$ | $100.00 \pm 0.00$ |
| 3.13 | $92.28 \pm 5.65$ | $91.98 \pm 1.17$ | $99.31 \pm 0.42$ |
| 6.25 | $66.29 \pm 4.16$ | $92.21 \pm 0.70$ | $103.07 \pm 1.39$ |
| 12.50 | $48.27 \pm 5.02$ | $93.84 \pm 2.51$ | $102.40 \pm 0.61$ |
| 25.00 | $43.11 \pm 3.80$ | $90.78 \pm 2.66$ | $106.84 \pm 3.81$ |
| 50.00 | $40.61 \pm 4.86$ | $95.19 \pm 1.17$ | $109.46 \pm 5.77$ |
| 100.00 | $43.47 \pm 4.69$ | $98.66 \pm 2.06$ | $111.44 \pm 4.29$ |
| 200.00 | $45.83 \pm 4.15$ | $94.19 \pm 2.45$ | $104.25 \pm 1.84$ |
| 500.00 | $28.92 \pm 1.42$ | $29.64 \pm 0.88$ | $83.59 \pm 2.76$ |

**Notes.**

Cells ($5 \times 10^3$ cells/well) were treated with gallic acid, BLE and BSE for 48 h before subjected to MTT assay. Results are expressed as means $\pm$ SEM.

BLE, Leaf water extract of *B. racemosa*; BSE, Stem water extract of *B. racemosa*; GA, Gallic acid.

(*Yeum et al., 2004*). These preliminary analyses showed that the *B. racemosa* extracts have very low toxicity and are only cytotoxic at very high concentrations (>200 µg/ml), which are not physiologically achievable.

In contrast, increasing concentrations of gallic acid was toxic to HepG2 cells whereby the concentration that inhibited 50% of cell proliferation ($IC_{50}$) was calculated as 11.6 µg/ml or 68 µM. Pure gallic acid was cytotoxic at high concentrations and its reported pro-oxidant activities could have caused the cell death, possibly by activating the Fenton reactions, leading to generation of $H_2O_2$ (*Kobayashi et al., 2004*). The pro-oxidant activity of gallic acid was also reported in a study using Caco-2 human colon and F344 rat liver cells (*Lee et al., 2005*). However, pure gallic acid at concentration less than 4.3 µg/ml or 25 µM was non-cytotoxic, with cell viability more than 80%.

Since the plant extracts was found to be non-cytotoxic, determination of its cellular antioxidant effects was conducted using two levels of concentrations, i.e., low concentrations (0.5, 1, 2 µg/ml) and high concentrations (5, 10, 20 µg/ml). Low concentrations were used to ascertain if changes in antioxidant responses could be seen at these concentrations.

## Cellular antioxidant status

The antioxidant status of HepG2 cells treated with BLE, BSE and gallic acid was measured using FRAP assay to evaluate the ferric reducing power of the cell lysate (Figs. 2A–2C). Treatment of HepG2 cells with gallic acid, BLE and BSE demonstrated increase in the antioxidant status of the cells compared to the control cells. The antioxidant activities of the treated cells did not show a dose-dependent relationship and the highest ferric reducing power was seen at a concentration of 1 µg/ml for the plant extracts and 1 µM for gallic acid. At this concentration, BLE showed a higher FRAP value than BSE and gallic acid. The presence of a variety of polyphenols in BLE as oppose to BSE could have contributed to the higher antioxidant activity. Moreover, the mixture of polyphenols in BLE as opposed

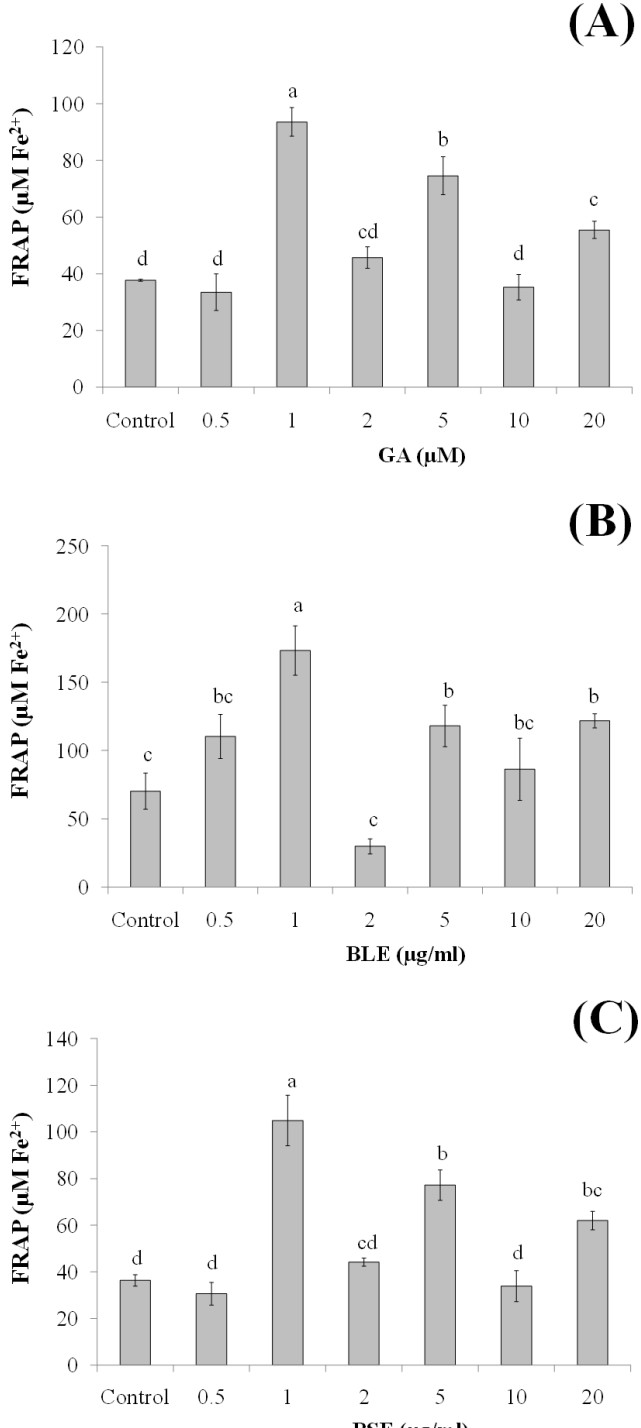

**Figure 2** **The effects of (A) gallic acid, (B) BLE and (C) BSE on antioxidant status of HepG2 cells.** Cells $(5 \times 10^3$ cells/well) were treated with gallic acid, BLE and BSE for 24 h and cellular antioxidant status was measured using FRAP assay. Results are expressed as means $\pm$ SEM. Values with different letters are significantly different at $p < 0.05$. BLE, Leaf water extract of *B. racemosa*; BSE, Stem water extract of *B. racemosa*; GA, Gallic acid; Control, untreated cells.
to pure gallic acid alone could indicate potential synergistic effects of the polyphenols in conferring the antioxidant effects.

The plasma concentration of polyphenols is relatively low, about 0.001–6 μM, due to their extensive metabolism (*Boulton, Walle & Walle, 1998*; *Spencer et al., 2008*). In a human bioavailability study, following the oral administration of gallic acid, its plasma level increased to 2 μM (*Shahrzad et al., 2001*). This concentration is slightly higher than the concentrations of gallic acid (1 μM) and the plant extracts (1 μg/ml) in our study in which high antioxidant activity was observed. This indicates that physiological concentration of the plant extracts was adequate to induce antioxidant protection. Furthermore, higher concentration of plant extracts may introduce xenobiotic stress to the cells (*D'Archivio et al., 2010*). The improved antioxidant status of the treated-cells in this study indicated the ability of exogenous antioxidants from *B. racemosa* to protect HepG2 cells against oxidative stress.

### Cytoprotective effects

This assay was conducted to ascertain the ability of gallic acid and the plant extracts to protect the HepG2 cells against cell death following induction of oxidative damage. Treatment of HepG2 cells with gallic acid (1 and 5 μM) and BLE and BSE (1 μg/ml), significantly protected the cells against $H_2O_2$-induced oxidative damage (Figs. 3A–3C). The increased antioxidant activities at this concentration, as measured by FRAP assay could have protected the cells against oxidative damage. However, increasing the concentrations of gallic acid, BLE and BSE did not further protect the cells from $H_2O_2$-induced oxidative damage. HepG2 cells treated with antioxidant-rich extracts such as olive oil, cocoa and common sage also improved antioxidant status of the cells and protected the cells against oxidative damage, further supporting the results from this study (*Goya, Mateos & Bravo, 2007*; *Lima et al., 2007*; *Martín et al., 2008*).

### Reactive oxygen species production

Measurement of ROS would give an indication on levels of oxidative stress. $H_2O_2$ was used as the source of ROS whereby $H_2O_2$ was converted to hydroxyl radicals and subsequently caused oxidation of dichlorodihydrofluorescein (DCFH) to dichlorofluorescein (DCF) complex, a fluorescent compound. In addition to hydroxyl radicals, other ROS including peroxyl radicals and lipid hydroperoxides can also contribute to formation of this fluorescent complex.

Pre-treatment of HepG2 cells with BLE and BSE prior to $H_2O_2$-induced oxidative stress gave lower fluorescent values compared to cells treated with $H_2O_2$ alone (Figs. 4B and 4C). Reduced fluorescence indicated that ROS production was reduced. Treatment of HepG2 cells with the plant extracts suppressed ROS production similar to the non-stressed cells. This implies that antioxidants in the extracts were able to inhibit ROS production and thus delay or prevent oxidative damage in the cells. In contrast to BLE and BSE, treatment of HepG2 cells with gallic acid only showed significant reduction in ROS production at concentrations above 2 μM although a reducing trend can be observed as the concentration of gallic acid increases (Fig. 4A). Gallic acid alone was not as effective as the plant extracts in reducing ROS production.

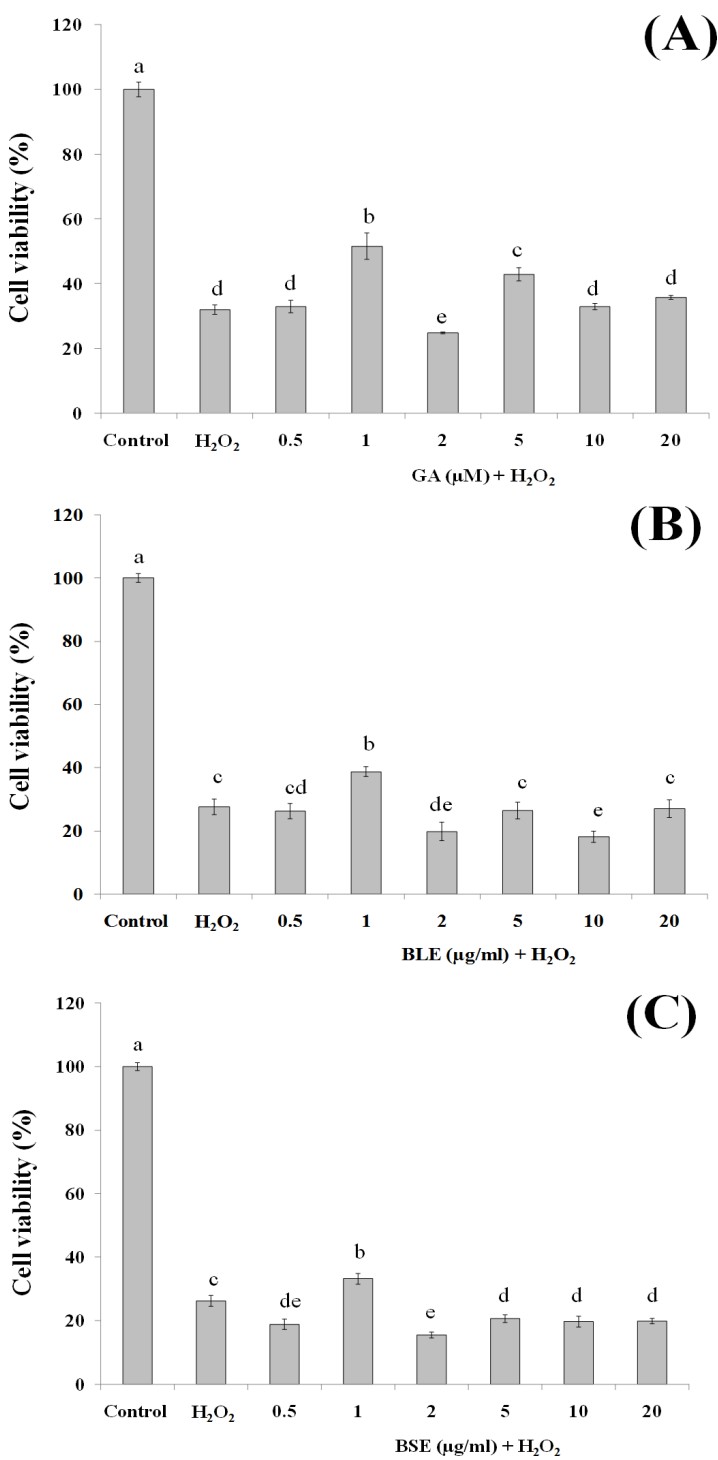

**Figure 3 The cytoprotective effects of (A) gallic acid, (B) BLE and (C) BSE on HepG2 cells following $H_2O_2$-induced oxidative damage.** Cells ($5 \times 10^3$ cells/well) were pre-treated with gallic acid, BLE and BSE for 24 h prior to $H_2O_2$-induced oxidative damage. Results are expressed as means $\pm$ SEM. Values with different letters are significantly different at $p < 0.05$. BLE, Leaf water extract of *B. racemosa*; BSE, Stem water extract of *B. racemosa*; GA, Gallic acid.

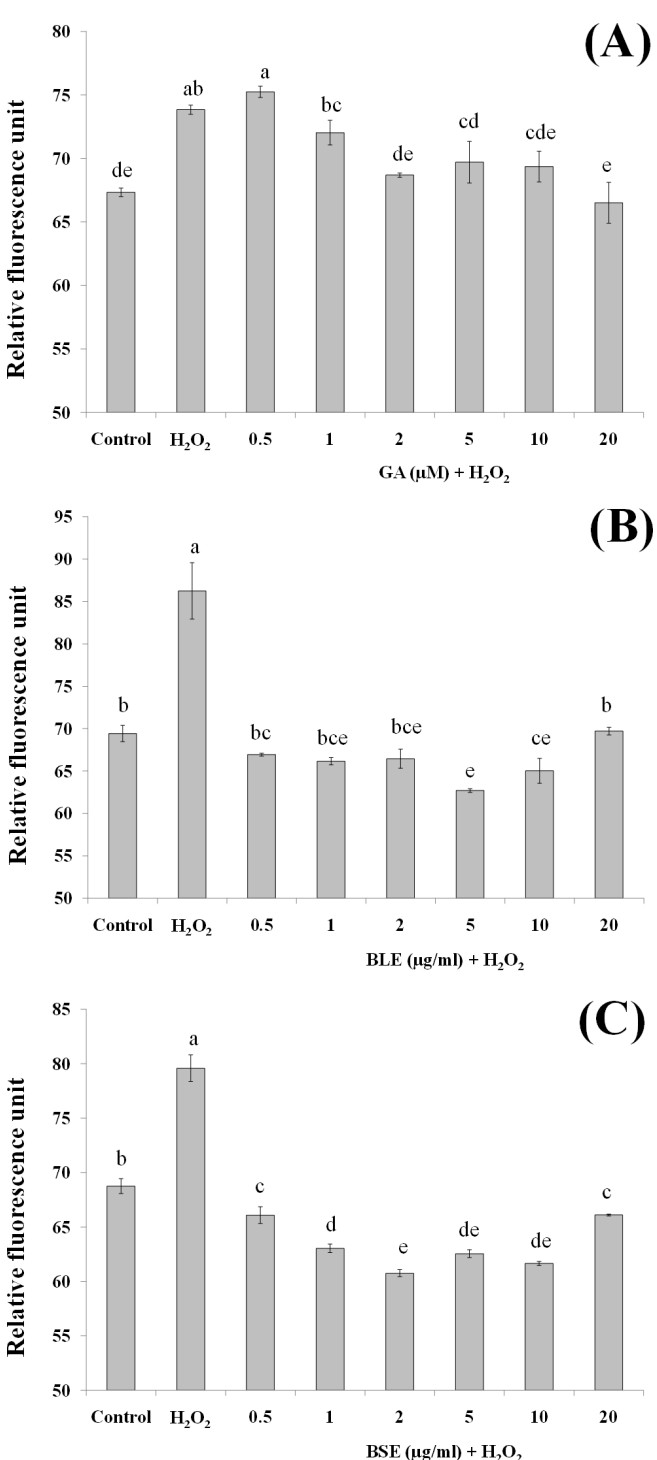

**Figure 4** **The effects of (A) gallic acid, (B) BLE and (C) BSE on ROS production of HepG2 cells following H₂O₂-induced oxidative damage.** Cells ($5 \times 10^3$ cells/well) were pre-treated with the plant extracts or gallic acid for 24 h prior to $H_2O_2$-induced oxidation. ROS production was determined by measuring relative fluorescence, using DCFH-DA probe. Values with different letters are significantly different at $p < 0.05$. BLE, Leaf water extract of *B. racemosa*; BSE, Stem water extract of *B. racemosa*; GA, Gallic acid; Control, negative control; $H_2O_2$, positive control.

This study demonstrates the potential synergistic effect of polyphenols in *B. racemosa* extracts in reducing oxidative damage as opposed to using a single bioactive compound. The polyphenols in BLE and BSE comprise of a mixture of polar phenolic acids to semipolar flavonoids. Due to the nature of their varying polarity, these polyphenolic antioxidants are able to react at the hydrophilic and hydrophobic phases of the cells to eliminate ROS (*Yeum et al., 2004*). Additionally, mutual synergistic effects of different polyphenolic compounds can enhance the antioxidative effect (*Dai & Mumper, 2010*).

## Analysis of lipid peroxidation

Since lipids in cell membrane are prone to oxidation, the effects of BLE and BSE in protecting against lipid peroxidation were also investigated. Lipids, especially polyunsaturated fatty acids (PUFA) at the membrane are susceptible to oxidative damage by ROS, forming lipid hydroperoxides and subsequently MDA (*Martín et al., 2008*), the latter being a widely used biomarker for oxidative stress (*Martín et al., 2010*).

Figures 5A–5C shows the MDA levels of the different treatment groups. HepG2 cells treated with $H_2O_2$ alone evoked a significant increase in the MDA levels, approximately three folds higher than the negative control containing medium alone. HepG2 cells treated with gallic acid, BLE and BSE showed significant reduction ($p < 0.05$) in MDA levels compared to positive control, indicating the ability of the samples to protect the cells against $H_2O_2$-induced lipid peroxidation. Results from this analysis also showed that low concentration of gallic acid and the plant extracts were adequate to prevent lipid peroxidation and that increasing the concentration of the extracts did not necessarily lead to higher inhibition of lipid peroxidation.

Polyphenols including gallic acid, quercetin and kaempferol that were detected in the extracts are strong scavengers of hydroxyl radicals (*Carocho & Ferreira, 2013*). Gallic acid was also able to protect liver cells, *in vitro*, against oxidative damage (*Senevirathne et al., 2012*). Previous studies reported that pre-incubation of HepG2 cells with rutin and quercetin could reduce lipid peroxidation (*Alía et al., 2006b*). Indeed, studies utilising polyphenolic-rich extracts such as purple sweet potato and common sage reported reduced lipid peroxidation in HepG2 cells, indicating the important roles of antioxidant polyphenols in providing protection against oxidative damage (*Hwang et al., 2011*; *Lima et al., 2007*).

## Activities of antioxidant enzymes

In addition to the direct effects of antioxidants in *B. racemosa* in scavenging ROS, bioactive compounds in the plant could protect against oxidative damage by influencing activities of antioxidant enzymes. Antioxidant enzymes play a vital role in modulating the redox balance of cells especially during oxidative stress. Changes in antioxidant enzyme activities is a fairly sensitive indicator of oxidative stress and can also be used to predict responses of antioxidants in plants (*Martín et al., 2008*). In this study, the activities of two major antioxidant enzymes; SOD and CAT were measured. SOD catalyses the dismutation of superoxide anion radicals ($O_2^{-\bullet}$) to produce $O_2$ and $H_2O_2$ (*Pieme et al., 2010*) whereas CAT catalyses the transformation of $H_2O_2$ to $H_2O$ (*Alía et al., 2006b*).

Treatment of HepG2 cells with $H_2O_2$ induced significant increase in the activities of SOD and CAT compared to control cells without $H_2O_2$-induced oxidation (Figs. 6A–6F).

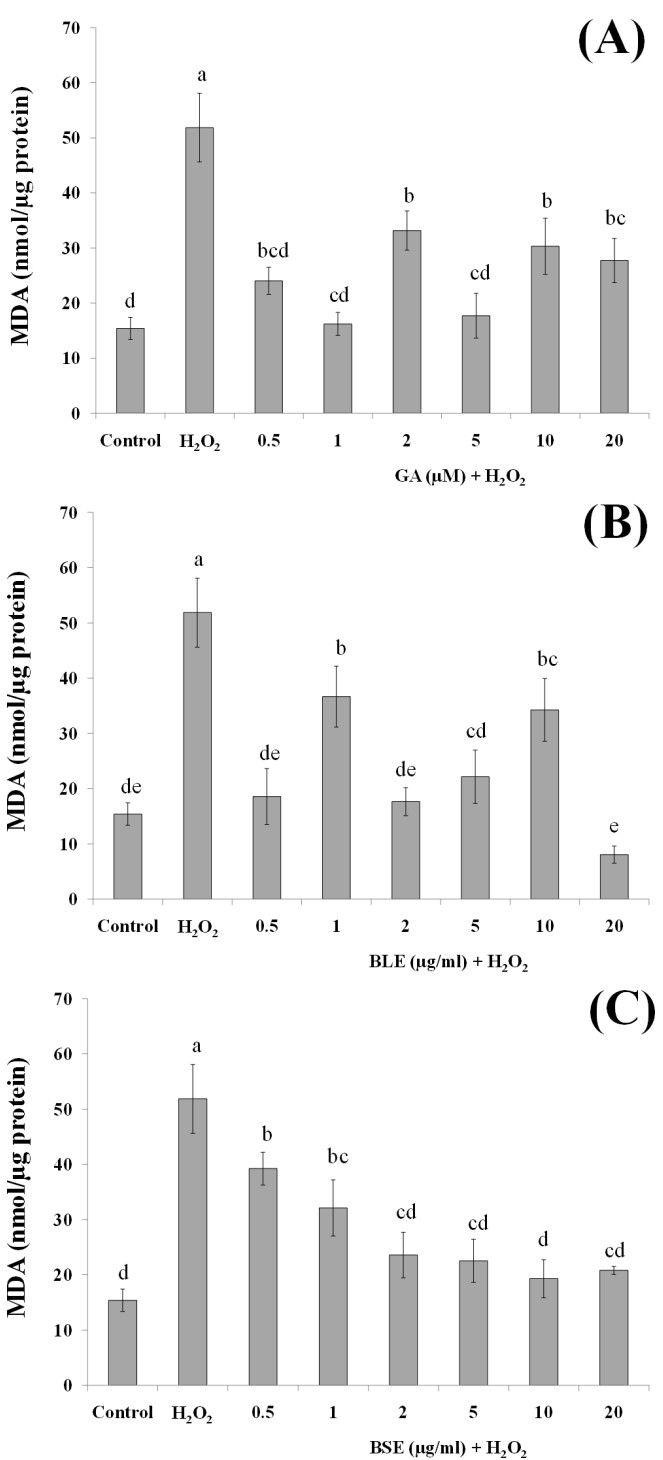

**Figure 5** **The effects of (A) gallic acid, (B) BLE and (C) BSE on lipid peroxidation of HepG2 cells following $H_2O_2$-induced oxidative damage.** Cells ($1.5 \times 10^5$ cells/well) were pre-treated with the plant extracts or gallic acid for 24 h prior to $H_2O_2$-induced oxidation. MDA was measured by the TBARS method. Values with different letters are significantly different at $p < 0.05$. BLE, Leaf water extract of *B. racemosa*; BSE, Stem water extract of *B. racemosa*; GA: Gallic acid; Control: negative control; $H_2O_2$, positive control.

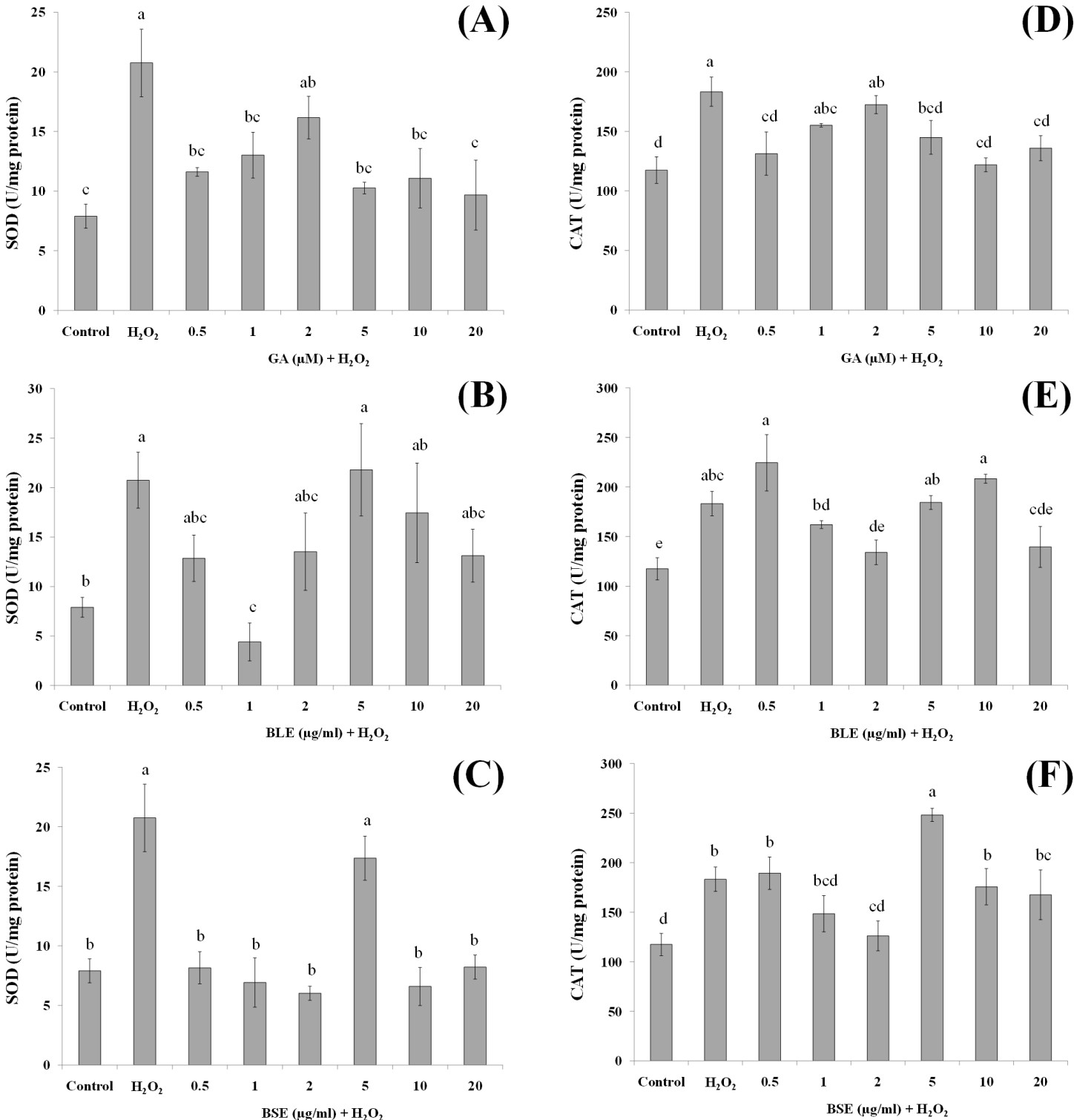

**Figure 6** **The effects of gallic acid, BLE and BSE on activities of SOD (A–C) and CAT (D–F) in HepG2 cells following $H_2O_2$-induced oxidative damage.** Cells ($1.5 \times 10^5$ cells/well) were pre-treated with the plant extracts or gallic acid for 24 h prior to induction of oxidation with $H_2O_2$. Results are expressed as means ± SEM. Values with different letters are significantly different at $p < 0.05$. BLE, Leaf water extract of *B. racemosa*; BSE, Stem water extract of *B. racemosa*; GA, Gallic acid.

Pre-treatment of HepG2 cells with 1 $\mu$g/ml BLE significantly reduced SOD activity by 79%. Although the remaining concentrations showed a reduced trend in SOD activity, this was not significant. BSE on the other hand, caused significant decrease in SOD activity (60–71%) at all tested concentrations except 5 $\mu$g/ml. Gallic acid reduced SOD activity significantly by 37–53% at all concentrations except 2 $\mu$M.

Similar to SOD, positive control cells with $H_2O_2$-induced oxidation showed higher activities of CAT than negative control cells without $H_2O_2$-induced oxidation. Pre-treatment with gallic acid at 0.5 $\mu$M and 5–20 $\mu$M significantly suppressed the activities of CAT by 20–30% in cells subjected to $H_2O_2$-induced oxidation. BLE, at 2 and 20 $\mu$g/ml significantly reduced 23–26% of CAT activity whereas a 31% decrease in CAT activity was observed in cells treated with 2 $\mu$g/ml BSE.

Positive control or cells treated only with $H_2O_2$ showed elevation of SOD and CAT activities, indicating a positive response of the cells in adapting towards increased production of ROS (*Martín et al., 2010*). The actions of SOD and CAT are closely related, whereby SOD reacts with $O_2^{-\bullet}$ to produce $H_2O_2$ that is subsequently reacted upon by CAT. Pre-treatment of the cells with gallic acid, BLE and BSE prior to induction of oxidative stress, led to reduced activities of SOD and CAT. Although in some instances, these reductions were not statistically significant, a reduced trend was observed. Epicatechin, quercetin and phenolic-rich cranberry powders were reported to prevent the increment of antioxidant enzyme activities during oxidative stress (*Alía et al., 2006b*; *Martín et al., 2010*; *Martín et al., 2015*). The ability of the *B. racemosa* extracts to regulate the activities of SOD and CAT indicate the potential of these extracts to assist the cells defense mechanism in responding towards oxidative stress.

## CONCLUSIONS

BLE and BSE at non-cytotoxic levels protected HepG2 cells against oxidative damage by acting as antioxidants, thus inhibiting ROS production and lipid peroxidation. In addition, the plant extracts also suppressed activities of the antioxidant enzymes SOD and CAT under conditions of oxidative stress. This current study indicates the potential use of the shoots of *B. racemosa* and its bioactive ingredients for the development of functional foods. Its antioxidant properties could provide the added ability to increase the antioxidant defense mechanism and to provide protection against oxidative stress-related diseases.

### Funding
This research project was funded by University of Malaya Research Grants (RG458/12HTM, FP015-2013B) and High-Impact Research Grant (H-20001-00-E000009) from University of Malaya, Kuala Lumpur, Malaysia. The funders had no role in study design, data collection and analysis, decision to publish, or preparation of the manuscript.

## Grant Disclosures

The following grant information was disclosed by the authors:
University of Malaya Research Grants: RG458/12HTM, FP015-2013B.
High-Impact Research Grant: H-20001-00-E000009.

## Competing Interests

The authors declare there are no competing interests.

## Author Contributions

- Kin Weng Kong conceived and designed the experiments, performed the experiments, analyzed the data, wrote the paper, prepared figures and/or tables, reviewed drafts of the paper.
- Sarni Mat-Junit and Azlina Abdul Aziz conceived and designed the experiments, analyzed the data, contributed reagents/materials/analysis tools, wrote the paper, reviewed drafts of the paper.
- Norhaniza Aminudin contributed reagents/materials/analysis tools, reviewed drafts of the paper.
- Fouad Abdulrahman Hassan performed the experiments.
- Amin Ismail contributed reagents/materials/analysis tools.

## Data Availability

The raw data is provided in Supplemental Information.

## Supplemental Information

Supplemental information for this article can be found online at http://dx.doi.org/10.7717/peerj.1628#supplemental-information.

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
