# Peer review of "Protective effects of the extracts of Barringtonia racemosa shoots against oxidative damage in HepG2 cells"

_PeerJ, doi:10.7717/peerj.1628_

## Round 0.1 · original submission · Major Revisions

In addition to the comments of the 2 reviewers, I have also reviewed the manuscript (see comments below) and these comments should also be considered if submission of a revised manuscript is made.

The manuscript describes the protective effects of Barrigntonia racemosa against induced oxidative damage in HepG2 cells. Through all the manuscript the authors describe that this aqueous plant extract exert a clear biological effect in vitro against oxidative damage in a cell system. Although the manuscript is well conducted and clearly written, the results are expected. Most of the effects showed here have been previously described for most of the pure polyphenols presented in this plant extracts, which clearly reduce the novelty of their findings.

Main comment: How do the authors explain the variability in response to FRAP analysis (Figure 3A-C)? This reviewer understand that the extract can have a maximum effect at 1 ug/mL, but how is possible that from that concentration the antioxidant effect varies randomly? Was all the range of concentrations performed at the same time? Please explain.

Minor comments:
P8L167 please indicate the plate format used
P9L195 please indicate the plate format used
Synergic effect of polyphenols have been previously described in the literature, unless it is experimentally shown in your results please reduce the tone by using “might” or “could” to point this possibility (P13L289).

Reviewer 1 ·

Basic reporting

The article fullfills the standards required. The work is clearly written and fits into the broader field of knowledge.
Just some minor revisions:
Page 1, Line 70 : Consider changing the phrase to "Hepatocellular carcinoma cells, HepG2, are a well established cell line"
Page 1, Line 72: Phenolic acids and flavonoids from plants are metabolised by the liver after absorption, mainly, in the small intestine.

Experimental design

The article fullfills the standards required. The primary research is original and fits within the scope of the journal. Methodology seems correct and reproducible.

Validity of the findings

The article fullfills the standards required. Data seems solid and statistically correct. Conclusions are appropriately stated, connected to the original question investigated, and are limited to those supported by the results.

Reviewer 2 ·

Basic reporting

Title: Concise and reflect the content of the article. Suggestions: The authors work with extracts and should be reflected in the title.

Abstract:
It is brief and describes clearly the purpose of the work and the major results.

Introduction:
This section is clear and well organised. Reflect the importance of using natural antioxidants to balance the oxidative stress to avoid aging and degenerative diseases. Specific improvements:
- L57 – Exogenous and endogenous antioxidants can limit or reduce the oxidative stress but no its elimination. Please, revise this paragraph.

Tables and figures:
Comments and corrections:
Figure2. These data could be more conclusive represented in a table where it will be easily to show the sample concentrations and the % of viability.

References:
The reference section is appropriate and updated. Corrections:
- L430 – Correct the surname of the author: D’Archivio
- L474 – This reference does not appear in the main manuscript.

Experimental design

Material and methods:
Properly describes the methodology used. Comments and corrections:
- L84 – Please, insert the nationality of the supply company
- L99 – Why do the authors subject the lyophilised extracts to acid hydrolysis? Please, explain it in this section. Did the authors analyse the samples without previous hydrolysis? The hydrolysis could degrade some polyphenols present in the extracts and the authors do not identify them but they are present in the in vitro assay.
- L101 – 90ºC
- L107 – 30ºC
- L110 – Please, substitute diode array detector for DAD.
- L110 – Detection at 280 nm is most commonly used for phenolics acids, although monitoring at 254 and 320 nm can provide more information about other polyphenols presents in the samples. Why do the authors use only the 280 nm wavelength?
- L127 – 37ºC
- L135 – MTT reagent was dissolve in?
- L146 – Why do the authors use these concentrations of gallic acid? No quantification of phenolic compounds in the samples was done.
- L150 – 37ºC
- L151 – Please, insert the absorbance reader specifications.
- L163 – Please, specify the expression of the results.
- L167 – Please, indicate the format of plate used for to seed the cells.
- L174 - Please, specify the expression of the results.
- L187 – 25 N
- L188 - 37ºC
- L195 - Please, indicate the format of plate used for to seed the cells.

Validity of the findings

Results and discussion: The presentation of the results and the discussion is clear, well-structured and follows a logical sequence.

Additional comments

PeerJ
Manuscript Number: Peerj-6893
Title: “Protective effects of Barringtonia racemosa shoots against oxidative damage in HepG2 cells”
Author (s): Kin Weng Kong, Sarni Mat-Junit, Norhaniza Aminudin, Fouad Abdulrahman Hassan, Amin Ismail, Azlina Abdul-Aziz

General comments
The search for naturally occurring compounds with beneficial health effects that can be used as potential food ingredients has a great interest. Therefore, the researches trying to characterize the potential active compounds from natural source and to associate some beneficial effects (antioxidant, anti-inflammatory, antitumor, antibacterial, etc) are very promising. This paper attempts to characterize natural plant extracts and associate them with certain beneficial properties such as its potential antioxidant capacity. The authors have recently studied the antioxidant activities of Barringtonia racemosa shoots (Kong et al., 2012 and 2014) as well as other authors (Razab et al., 2010), but in this novel work, the authors try to proof the antioxidant activity in vitro using human cells. Therefore, the objective addressed in the present article can be considered of great interest and meets the requirements for scientific publication.

---

## Round 0.2 · accepted · Accept

The manuscript is acceptable in its present form.